# The Impact of Changes in the Intake of Fiber and Antioxidants on the Development of Chronic Obstructive Pulmonary Disease

**DOI:** 10.3390/nu13020580

**Published:** 2021-02-10

**Authors:** Young Ju Jung, Se Hee Lee, Ji Ho Chang, Hye Seung Lee, Eun Hee Kang, Sei Won Lee

**Affiliations:** 1Department of Pulmonology and Critical Care Medicine, Asan Medical Center, University of Ulsan College of Medicine, Seoul 05505, Korea; tazo76@amc.seoul.kr; 2Health Screening and Promotion Center, Asan Medical Center, Seoul 05505, Korea; jhchang@amc.seoul.kr (J.H.C.); grayhs@nate.com (H.S.L.); ehkang@amc.seoul.kr (E.H.K.); 3Department of Pulmonology, Allergy and Critical Care Medicine, CHA Bundang Medical Center, CHA University, Seongnam 13496, Korea; celestia7@gmail.com

**Keywords:** dietary fiber, antioxidants, diet, lung function, chronic obstructive pulmonary disease

## Abstract

Diet is a health-related factor that can modify lung function. This study hypothesized that the change in age-related dietary intake affects lung function. The subjects who undertook a dietary assessment and spirometry in 2012 and 2017, were retrospectively collected in a health screening center. Dietary intakes were directly evaluated using food frequency questionnaires (FFQ) administered by trained dietitians and were compared at the baseline (2012) and 5-year follow-up (2017). A forced expiratory volume in one second (FEV_1_)/forced vital capacity (FVC) value below 0.70 was defined as airflow limitation. Logistic regression models were used to estimate the odds ratio (ORs) adjusted for potential confounders. A total of 1439 subjects with normal spirometry were enrolled. New airflow limitations were detected in 48 subjects (3.3%) at the 5-year follow-up, including 41 (85.4%) men and 11 (22.9%) current smokers. After adjusting for age, sex, smoking history, and baseline FEV1/FVC, the odd ratios (OR) for new airflow limitation in fiber, vitamin C, and folic acid per 10% decrease in daily recommended requirement were 2.714 (95% confidence interval (CI), 1.538–4.807; *p* = 0.001), 1.083 (95% CI: 1.020–1.149; *p* = 0.007), and 1.495 (95% CI: 1.172–1.913; *p* = 0.001), respectively. A decreased intake of dietary fiber, vitamin C, and folic acid is associated with a newly developed airflow limitation.

## 1. Introduction

Chronic obstructive pulmonary disease (COPD), characterized as persistent airflow limitation, is a major global health concern with a prevalence of 5–25% in adults and the fourth leading cause of death worldwide [1,2]. COPD is treatable but is a progressive disease that generally cannot be cured. Smoking is the most important risk factor for COPD development and progression but it is well-established that its effect is individually different. Clinically significant airflow limitations never develop in 50–85% of smokers [3,4]. This indicates that there are ‘susceptible’ and ‘non-susceptible’ phenotypes among smokers [5] and the factors that decide this are of considerable interest. In terms of innate factors that contribute to COPD, multiple loci have been identified in genome-wide association studies [6] and these cannot be modified. In relation to acquired factors, physical activity is suggested to prevent COPD development [7]. However, COPD still progresses in some cases even after smoking cessation and appropriate physical activity. Hence, additional modifiable factors are likely to contribute to COPD and need to be identified to improve patient management.

Diet is one of the most important lifestyle determinants of health but is relatively ignored in relation to lung disorders. Current dietary guidelines do not provide enough information in this regard [2] and it is therefore not easy for physicians to provide appropriate guidance about a beneficial diet for lung health. Some dietary evidence is available from prior reports, however. Cross-sectional studies have reported a negative association between fruit, vegetable, and vitamin C intakes and lung function decline [8]. Moreover, a diet that includes fruits, vegetables, fish, whole grains, and oily fish has been shown to be associated with a lower risk of COPD development [9,10,11]. Dietary fiber intake has also been associated with better lung function and a lower prevalence of COPD [12]. There have not been many longitudinal studies of this relationship but one such study has indicated that a high fiber diet, especially cereal fiber, can help to reduce the risk of COPD development [13]. Most of these prior studies were performed in Europe and North America however, limiting the general applicability of the findings [14]. Diets vary across geographical regions and the main sources of nutrients also differ. Furthermore, prior studies have not considered the dynamic aspects of individual changes in the diet with regard to lung function. Changes in the dietary intakes may be also important, considering that meeting the daily micronutrient requirement becomes more difficult with age.

We hypothesized that the dietary intake includes dynamic changes that would have an impact on the airflow limitation, which is the principal feature of COPD, and each nutrient might be associated with it. In our current study therefore, we analyzed the dietary patterns and lung function in a large sample population that undertook the same detailed nutrition assessment and pulmonary function tests with an interval of five years. We focused on the relationship between new airflow limitation development and changes to the dietary pattern.

## 2. Methods

### 2.1. Study Subjects

The study subjects were retrospectively screened from 1832 cases who underwent medical check-ups including spirometry and detailed dietary assessment at a disease-screening private facility (Health Promotion Center, Asan Medical Center, Seoul, Korea) in 2012 and 2017. For our analyses, we selected 1745 subjects with spirometry data and available food frequency questionnaire (FFQ), excluding any subjects without FFQ (*n* = 78) or spirometry (*n* = 9) data. Based on the results of normal spirometry at the baseline year in 2012, we determined whether there were any new cases in this population of airflow limitation in 2017. We thus enrolled 1439 adults with normal spirometry in 2012 with an available baseline FFQ. A flowchart of the subjects included in this study is provided in Figure 1. Our institutional review board approved this retrospective study and waived the requirement for informed written consent (IRB no. 2019-0061).

### 2.2. Assessment of Dietary Intake

Dietary habits during the prior month were assessed using seven questions such as the frequency and regularity of meals, breakfast skipping, eating speed, and the frequency of overeating, eating-out, and caffeine intake. These self-reported questionnaires were collected before the health screening. Dietary intake information over the previous 3 months was assessed using a 117-item food frequency questionnaire by a trained dietitian [15]. The consumption frequency of each food item was classified into nine categories: never or seldom, once a month, 2–3 times a month, 1–2 times a week, 3–4 times a week, 5–6 times a week, once a day, twice a day, or 3 times or more every day. Standard portion sizes were listed with each food type, and all food groups were compared with the dietary reference intakes for Koreans (KDRI) [16], which indicate the recommended daily intake. All dietary intakes and changes were compared between baseline (2012) and 5-year follow-up (2017).

### 2.3. Assessment of Baseline Health Status and Laboratory Data

Self-administered health questionnaire responses were obtained including demographic data and medical history, particularly respiratory disease. Each subject completed a questionnaire concerning respiratory symptoms and smoking history. Physical body measurements such as height and weight, and laboratory tests for white blood cell (WBC) counts were included in the health check-up.

### 2.4. Spirometry

Spirometry was conducted by an experienced laboratory technologist using a VMax 20 spirometer (Viasys, San Diego, CA, USA), in accordance with American Thoracic Society recommendations [17]. Predicted spirometry values were calculated from Korean reference equations that are based on representative samples from the Korean population [18]. The criteria used for normal spirometry findings were a pre-bronchodilator FEV_1_/FVC ratio ≥0.70 and an FVC and FEV_1_ ≥80% predicted—FVC, forced vital capacity; FEV_1_, forced expiratory volume in one second. An FEV_1_/FVC ratio <0.70 was defined as airflow limitation.

### 2.5. Statistical Analysis 

All analyses were performed using SPSS software, v21.0 (SPSS Inc., Chicago, IL, USA). Between-group comparisons were made using the t-test and analysis of variance for continuous variables and Chi-squared analysis for categorical data. All tests for significance were two-sided, and *p*-values of <0.05 were considered to indicate significance. Univariate and multivariate regression models were used to evaluate the association between dietary intake changes from baseline to 5-year follow-up and newly developed airflow limitations. Multivariate logistic regression models were used to estimate odds ratios (ORs) adjusted for potential confounders including age, sex, smoking history, and FEV_1_/FVC ratio at baseline (2012).

## 3. Results

The mean ± standard error of the mean age of the subjects at baseline was 52.53 ± 0.21 years. The mean body mass index was 23.3 kg/m2, and 517 (35.9%) of the 1439 subjects were male. At baseline, 976 (67.8%) of the study subjects had never been smokers, 278 (19.3%) were former smokers, and 185 (12.9%) were current smokers. These overall characteristics were similar between 2012 and 2017 in relation to smoking history, respiratory disease history, and respiratory symptoms, but 48 subjects (3.3%) showed new airflow limitations in 2017. The individuals with an airflow limitation were more likely to be older (64.10 ± 1.19 vs. 57.28 ± 0.21 years, *p* < 0.001) and male (85.42% vs. 34.22%, *p* < 0.001) and had a significantly higher smoking history, respiratory disease history, and respiratory symptoms. As expected, the FEV_1_ (84.67% vs. 94.28% of predicted value, *p* < 0.001) and FEV_1_/FVC (66.25% vs. 79.18%, *p* < 0.001) were lower in the group with an airflow limitation (Table 1).

Almost all dietary intakes except cholesterol had significantly decreased at the 5-year follow up compared with the baseline. Notably however, the decrease in fiber, folic acid, vitamin A, vitamin B1, vitamin B6, and vitamin C was significantly greater in the subjects showing a new airflow limitation (Table 2). Based on these results, we further investigated the possible associations between dietary composition changes and new airflow limitations. With adjustments for age, sex, smoking history, and FEV_1_/FVC ratio at baseline, the decrease of fiber, vitamin C, and folic acid showed a significant positive association with a new airflow limitation. The ORs of new airflow limitation were 1.527 (95% confidence interval (CI): 1.200–1.944; *p* = 0.001) for a 1 g/day decrease in fiber, 1.008 (95% CI: 1.002–1.014; *p* = 0.007) for a 1 mg/day decrease in vitamin C, and 1.010 (95% CI: 1.004–1.016; *p* = 0.001) for a 1 µg/day decrease in folic acid. With regard to a 10% decrease in daily recommended nutrient requirements (KDRI), fiber showed the highest OR of 2.714 (95% CI: 1.538–4.807) for fiber, followed by 1.083 (1.020–1.149) for vitamin C and 1.495 (1.172–1.913) for folic acid (Figure 2).

To further examine the effects of changes in dietary fiber, folic acid, and vitamin C, these intake and changes were divided into quartiles and then compared with the onset of new airflow limitations. The proportion of new airflow limitation cases increased significantly with a higher reduction in the dietary intake of fiber (1.39% vs. 5.85% for quartile 1 vs. quartile 4, *p* = 0.005) and folic acid (1.67% vs. 5.01% for quartile 1 vs. quartile 4, *p* = 0.050) 5 years after baseline (Figure 3). In cross-sectional analysis in 2017, we found that a higher dietary vitamin C intake was associated with a lower onset of airflow limitation at the 5-year follow up with marginal significance (5.29% vs. 1.67% for quartile 1 vs. quartile 4, *p* = 0.059, Appendix A). The WBC counts were lowest in the highest quartile of vitamin C intake (5.45 ± 0.09 vs. 5.18 ± 0.07 × 10^3^/ µL for quartile 1 vs. quartile 4, *p* = 0.006, Figure 4). The WBC counts were analyzed to assess the possible underlying mechanisms, such as the anti-inflammatory effects of micronutrients as they are closely related to the smoking status (Appendix A).

Smoking can affect airflow strongly and we thus assessed the effects of diet in combination with smoking history. The age was similar between the groups with and without a smoking history. Males were predominant among the former and current smokers than nonsmokers (89.5% vs. 11.2%, *p* < 0.001), and the mean pack years was 28.6 (Appendix A). Cross-sectional analysis of the subjects in 2017 showed a significantly lower intake of fiber and folic acid in the nonsmoker group with a new airflow limitation. In terms of the differences between baseline and 5-year follow-up among the nonsmokers, the decreases in the intakes of fiber, folic acid, and vitamin C were more prominent in those with airflow limitation than in those without (−3.72 ± 0.08 vs. −2.02 ± 0.06 g, *p* = 0.002: −137.07 ± 22.17 vs. −71.71 ± 2.34 µg, *p* = 0.003: −87.84 ± 2.98 vs. −35.78 ± 2.39 mg, *p* = 0.021, respectively). Although the mean intake of these nutrients also decreased in former and current smokers, the differences were not significant (Appendix A). WBC counts were lowest in the highest quartile of vitamin C intake among smokers (6.00 ± 0.16 vs. 5.49 ± 0.15 × 10^3^/µL for quartile 1 vs. quartile 4, *p* = 0.014, Appendix A).

Further analysis was performed to determine whether sex was a factor in the relationship between diet and airflow limitations. The proportion of ever-smokers was higher among the male subjects (78.92% vs. 6.19%, *p* < 0.001). In addition, more of the males in our study series had a history of chronic respiratory disease including COPD, tuberculosis, or asthma (9.09% vs. 5.86%, *p* = 0.021). WBC counts were slightly but significantly higher in the male subjects (5.72 ± 0.07 vs. 5.12 ± 0.05 × 10^3^/µL, *p* < 0.001) whereas the ratio of the predicted FEV_1_ was higher in the females (92.09 ± 0.41 vs. 95.01 ± 0.32%, *p* < 0.001) (Appendix A). In the male subjects, there was no significant association between any of the dietary intake components and new airflow limitations. However, the fiber and folic acid intakes were significantly lower in the females with new airflow limitations. In terms of the differences between baseline and the 5-year follow-up, fiber was the only significant component for which a decrease had an association with new airflow limitation in both male and female subjects (−2.80 ± 0.30 vs. −2.10 ± 0.07 g, *p* = 0.009 and −3.91 ± 0.28 vs. −2.03 ± 0.05 g, *p* = 0.004, respectively). Vitamin B1 showed a significant association in the male and folic acid in the female (Appendix A). In the male subjects, the WBC counts were lower in the highest quartile of dietary intakes than in the lowest quartile for fiber (6.02 ± 0.16 vs. 5.35 ± 0.12 × 10^3^/µL for quartile 1 vs. quartile 4, *p* = 0.008), and for vitamin C (6.01 ± 0.15 vs.5.40 ± 0.12 × 10^3^/µL for quartile 1 vs. quartile 4, *p* = 0.002, Appendix A). There was no significant relationship between the dietary intake and WBC counts in the female subjects.

## 4. Discussion

In our present analyses, the change in dietary intakes and their association with newly developed airflow limitations were analyzed for a relatively large number of health-checkup subjects with an initially normal lung function. The changes in the intake of dietary components including fiber, vitamin C, and folic acid showed a significant association with the development of airflow limitation when we adjusted for age, smoking history, and baseline pulmonary function. A higher decrease in these nutrients positively correlated with a higher onset of new airflow limitations over a 5-year period. The decrease in fiber intake in particular showed a consistent association in both the males and females in our current study cohort.

The importance of maintaining an adequate nutritional intake for better health and prevention of disease is well recognized. As a consequence of age-related physiological and functional changes, it can become more difficult for these nutritional needs to be met [19,20]. A decrease in micronutrient intake usually starts from the ages of 50 to 60 [21,22] and becomes particularly evident at an age above 75 years [23]. In our present study cohort, the intake levels of almost all nutrients except cholesterol were found to be significantly decreased at the 5-year follow up, as noted in other epidemiologic studies. This kind of dietary change has various effects on health. Inadequate folate intakes, which are common amongst older populations [24,25], are associated with increased plasma levels of homocysteine, an independent risk factor for heart disease and stroke [26]. Lower dietary fiber intake increases the risk of cardiovascular disease, type 2 diabetes, and colorectal cancer [27,28,29]. In the elderly population aged from 60–94, vitamin C has been shown to be the major dietary component that could be strongly linked to cognitive function [30]. The shortfall of micronutrient intake in the elderly causes adverse health consequences, combined with an energy dense but fiber or antioxidant deficient diet [23]. Until now, the impact of this shortfall on lung health was unclear. In our current study, lower fiber and lower vitamin intakes were more prominent in the group with new airflow limitations. In this context, our study makes the important observation that this decreased intake of micronutrients, usually associated with ageing, also has an adverse effect on lung function.

The epidemiologic evidence indicates in the main that vitamin C, fruits, and vegetables are beneficial for lung function [31]. Many studies have indicated this through an analysis of the relationship between diet and lung function. However, the dynamic aspects of diet intake, which can change with ageing, have been ignored. Our present findings were meaningful in this aspect, showing that the maintenance or increase in intake of fiber, vitamin C, and folic acid can contribute to the prevention of new airflow limitation caused by ageing- and smoking-related processes. There are various possible explanations for this relationship. In the first instance, these nutrients have known anti-inflammatory and antioxidant effects. In our present analyses, WBC counts, an inflammatory marker, were associated with reduced lung function [32,33], and were affected by the intake of fiber in some of the subjects. Fiber is fermented by intestinal microorganisms to produce various anti-inflammatory metabolites, such as short chain fatty acids (SCFAs) [34,35]. Systemic benefits that go beyond the intestine have been documented for fiber through SCFAs in chronic lung disease and asthma [36]. Vitamin B6 and folic acid are well-known antioxidants which can therefore reduce oxidative stress from air pollution or smoking. Higher levels of oxidative markers in COPD have been found to correlate with decreased lung function [37]. Oxidative stress persists long after smoking cessation as a result of the continuous production of pro-oxidants, which is more prominent in COPD [38]. Low serum antioxidant vitamin levels appear to increase the risk of airflow limitation associated with smoking exposure [39]. Moreover, smokers tend to choose an unhealthier diet than never- or ex-smokers [40]. Indeed, in our study, the smokers consumed less fiber, vitamin C, and folic acid.

It becomes more difficult to meet healthy dietary requirements with age and age-related decreases in antioxidant enzyme activity are reported to contribute to increased oxidative damage and age-related degeneration. Physiological and social changes such as decreased food intake, impaired sensory perception, malabsorption, declining activity, and increased disability increase the risk of inadequate nutritional status (particularly for micronutrients) in the older population group [38,41]. This suggests the need for a continuous or even increased supply of key antioxidant nutrients in older individuals. Fortified food may help to resolve this shortfall in nutritional intake. A previous ageing cohort study found that the biomarker status of vitamins B2, B6, B12 and folate significantly increased with a higher intake of fortified foods [42]. Another study showed a significant contribution of fortified foods to the higher intake of micronutrients in older adults [43]. It is not yet certain however that these fortified foods really help to prevent age-related disease beyond simply replenishing nutrition. The evidence for respiratory diseases is even more lacking. Further research is warranted to elucidate the effects of these intakes on chronic respiratory disease.

Based on our current data and the results of previous studies, the appropriate guidelines for dietary intake in patients with COPD or in populations vulnerable to chronic lung disease is an important issue to be resolved. Foods rich in fiber and antioxidants, typically fresh fruits and vegetables, are reported to be beneficial in relation to respiratory symptoms [44], lung function decline [45], COPD incidence [46], and mortality [47]. The importance of these nutrients to lung function has been further suggested by our present findings. Additionally, we also elucidated from our current data that a decreased intake of fiber and antioxidants with ageing adversely affects lung function. This is particularly important for older patients who are susceptible to lung decline and in which it is less easy to maintain nutritional intake. The hyperinflation in emphysema for example can cause flatulence which makes it more difficult to eat fruits or vegetables. Effective ways to increase or at least maintain the intake of essential micronutrients should thus be sought in patients with chronic respiratory disease. Small frequent meals that are dense in essential nutrients and that require little preparation, resting before meals, and taking daily doses of multivitamins are possible recommendations in this regard [48].

There were some limitations of our current study of note. In the first instance, a causal relationship between dietary intake and lung function remains to be definitively concluded. Dietary patterns can also be affected by various factors, such as health concerns, a patient’s economic situation, emotional stress, and cooking availability. To minimize for confounders in our present analysis, we corrected for demographics, baseline lung function, and smoking history. Evidence for an association with inflammatory markers supports the effect of diet on lung function. A future interventional study with detailed monitoring of lifestyles may provide clearer findings.

## 5. Conclusions

In conclusion, the intake of dietary fiber and antioxidants changes with ageing and their decreased intake are associated with the new onset of airflow limitations. Our present study suggests that adequate nutritional intake can play an important role in preserving lung function and preventing or delaying the progression of age-related diseases such as COPD. Maintaining the intake of fiber and vitamins in older people is also becoming an important issue in ageing societies.

## Figures and Tables

**Figure 1 nutrients-13-00580-f001:**
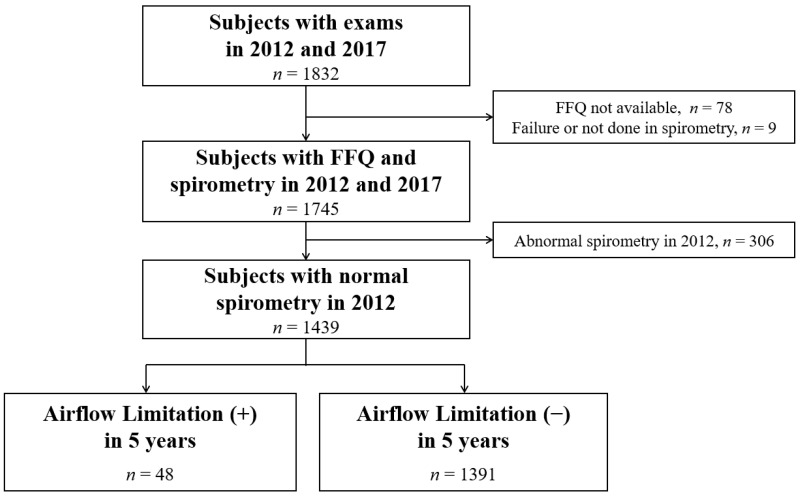
Flow diagram of the study population. A total of 1439 examined subjects with normal spirometry at baseline were enrolled among which 48 (3.3%) had developed a new airflow limitation after 5 years. FFQ, Food Frequency Questionnaire.

**Figure 2 nutrients-13-00580-f002:**
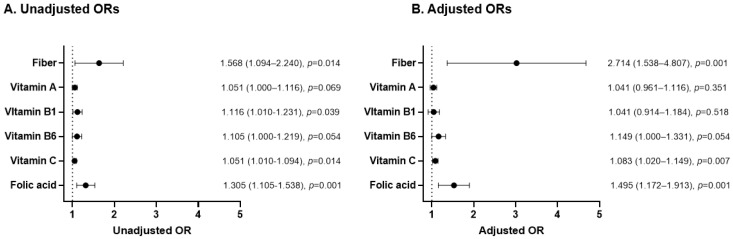
Association between new airflow limitation onset and nutrient intake differences between the baseline and 5-year follow-up. Adjusted for gender, age, smoking history, and FEV_1_/FVC at baseline (2012). (**A**) Unadjusted OR (**B**) Adjusted ORs is added to the end of Figure 2 title. All values are expressed with a 95% confidence interval. Only diet composition with statistical significance in univariate analysis was selected. OR, odds ratio.

**Figure 3 nutrients-13-00580-f003:**
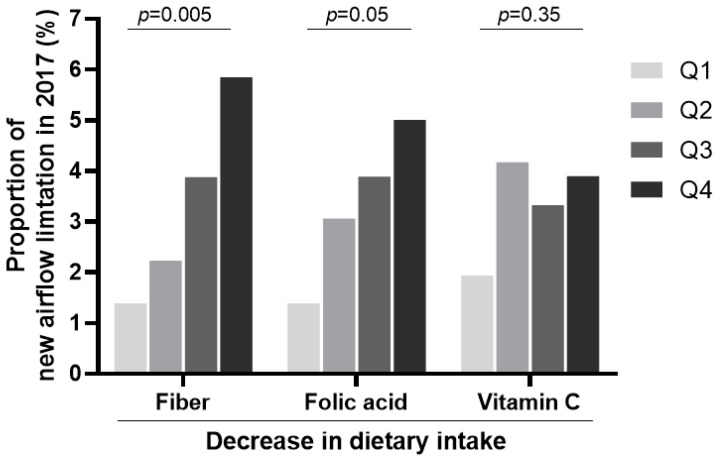
The proportion of subjects showing a new airflow limitation at the 5-year follow up in different nutrient intake quartiles. The development of new airflow limitation was highest in the quartile showing the highest decrease in fiber, folic acid, and vitamin C intake. The differences were significant for fiber and folic acid. The decreased intakes of each nutrient in the quartiles were as follows: fiber Q1, −5.515~0.999: fiber Q2, 1.000~2.049: fiber Q3, 2.050~3.190: fiber Q4, 3.191~7.770; folic acid Q1, −231.561~32.579: folic acid Q2, 32.580~81.903: folic acid Q3, 81.904~125.063: folic acid Q4, 125.064~330.884; vitamin C Q1, −305.291~3.305: vitamin C Q2, 3.306~38.078: vitamin C Q3, 38.079~74.457: vitamin C Q4, 74.458~325.694. Q, quartile.

**Figure 4 nutrients-13-00580-f004:**
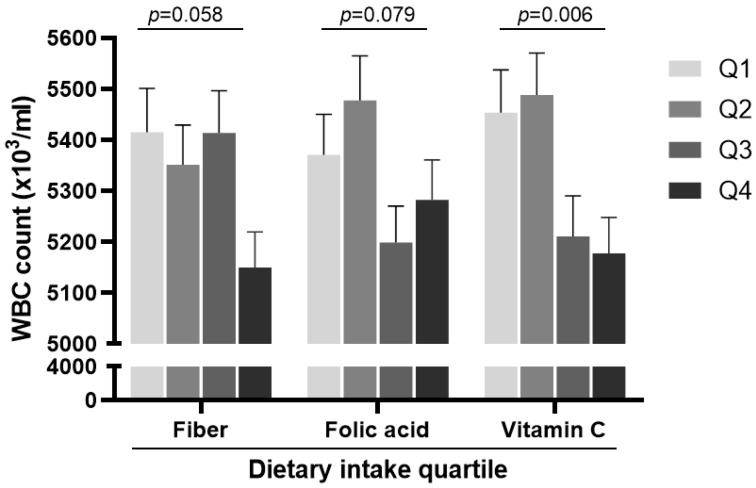
WBC counts in the different nutrient intake quartiles at the 5-year follow up. WBC, white blood cells.

**Table 1 nutrients-13-00580-t001:** Baseline characteristics of the study subjects according to the presence of a new airflow limitation at a 5-year follow-up.

Variables	2012	2017			
	Total *n* = 1439	Total *n* = 1439	Airflow Limitation (+) *n* = 48	Airflow Limitation (−) *n* = 1391	*p*-Value *
Males, *n* (%)	517 (35.93)	517(35.93)	41 (85.42)	476 (34.22)	<0.001
Age, years	52.53 (0.21)	57.51 (0.21)	64.10 (1.19)	57.28 (0.21)	<0.001
BMI, kg/m^3^, *n* (%)	23.26 (0.08)	23.27 (0.08)	24.38 (0.39)	23.23 (0.08)	0.008
AC, cm	81.23 (0.24)	82.46 (0.23)	87.75 (1.19)	82.25 (0.24)	<0.001
Smoking history, *n* (%)					<0.001
Nonsmoker	976 (67.82)	974 (67.69)	11 (22.92)	963 (69.23)	
Former smoker	278 (19.32)	314 (21.82)	26 (54.16)	288 (20.70)	
Current smoker	185 (12.86)	151(10.49)	11 (22.92)	140 (10.07)	
Pack years ^†^, years	27.72 (0.82)	28.86 (0.88))	42.07 (3.79)	27.56 (0.88)	0.001
History of respiratory disease, *n* (%)	89 (6.18)	101 (7.02)	10 (20.83)	91 (6.54)	<0.001
TB	69 (4.79)	71 (4.93)	6 (12.50)	65 (4.67)	0.014
COPD	2 (0.14)	4 (0.28)	2 (4.17)	2 (0.14)	0.006
Asthma	19 (1.32)	29 (2.02)	4 (8.33)	25 (1.80)	0.003
Respiratory symptom	512 (35.58)	537 (37.32)	26 (54.16)	511 (36.74)	0.014
DOE	442 (30.72)	447 (31.06)	20 (41.67)	427 (30.70)	0.106
Phlegm	64 (4.45)	65 (4.52)	4 (8.33)	61 (4.39)	0.272
Cough	58 (4.03)	52 (3.61)	2 (4.17)	50 (3.60)	0.691
Wheezing	36 (2.50)	38 (2.64)	3 (6.25)	35 (2.52)	0.130
Regular exercise, *n* (%)	871 (60.53)	925 (64.28)	36 (75.00)	889 (63.92)	0.116
Laboratory finding					
WBC (× 10^3^/µL)	5.34 (0.04)	5.33 (0.04)	5.78 (0.31)	5.32 (0.04)	0.145
CRP (mg/L)	0.09 (0.005)	0.10 (0.004)	0.13 (0.025)	0.10 (0.004)	0.170
Glucose (mg/dL)	96.80 (0.48)	102.84 (0.61)	107.31 (2.44)	102.28 (0.62)	0.072
LDL (mg/dL)	119.41 (0.81)	125.40 (0.92)	119.40 (4.96)	125.61 (0.93)	0.224
HDL (mg/dL)	58.54 (0.38)	61.03 (0.45)	55.69 (2.23)	61.21 (0.46)	0.027
TG (mg/dL)	109.65 (1.81)	109.40 (1.72)	122.02 (10.19)	108.97 (1.74)	0.172
Spirometry					
FVC (L)	3.58 (0.02)	3.51 (0.02)	4.15 (0.12)	3.49 (0.02)	0.001
FVC (% of predicted value)	94.18 (0.23)	93.74 (0.25)	93.79 (1.53)	93.74 (0.25)	0.972
FEV_1_ (L)	2.88 (0.01)	2.75 (0.01)	2.75 (0.08)	2.75 (0.01)	0.934
FEV_1_ (% of predicted value)	94.73 (0.23)	93.96 (0.25)	84.67 (1.56)	94.28 (0.25)	<0.001
FEV_1_/FVC	80.82 (0.13)	78.75 (0.14)	66.25 (0.35)	79.18 (0.13)	<0.001

* Compared between groups with and without a new airflow limitation. ^†^ Calculated among smokers. All values are expressed as a mean (standard error), unless otherwise stated. BMI, body mass index; AC, abdominal circumference; TB, tuberculosis; COPD, chronic obstructive pulmonary disease; DOE, dyspnea on exertion; WBC, white blood cell count; CRP, C-reactive protein; LDL, low density lipoprotein cholesterol; HDL, high density lipoprotein cholesterol; TG, triglyceride; FVC, forced vital capacity; FEV_1_, forced expiratory volume in one second.

**Table 2 nutrients-13-00580-t002:** Dietary fiber and other nutrient intake differences between 2012 and 2017 in the study population.

Nutrients	2012	2017				Difference between 2012 and 2017		
	Total *n* = 1439	Total *n* = 1439	Airflow Limitation (+) *n* = 48	Airflow Limitation (−) *n* = 1,391	*p*-Value *	Airflow Limitation (+) *n* = 48	Airflow Limitation (−) *n* = 1391	*p*-Value
Fiber, g	8.45 (0.04)	6.36 (0.04)	6.14 (0.23)	6.37 (0.04)	0.305	−2.97 (0.28)	−2.06 (0.05)	<0.001
Vitamin A, RE	1097.79 (7.33)	806.52 (7.23)	811.85 (41.63)	806.34 (7.34)	0.891	−392.87 (45.00)	−287.76 (8.91)	0.031
Vitamin B1, mg	1.15 (0.01)	0.97 (0.01)	1.03 (0.04)	0.97 (0.01)	0.113	−0.28 (0.05)	−0.18 (0.01)	0.021
Vitamin B2, mg	1.18 (0.01)	1.01 (0.01)	1.12 (0.06)	1.01 (0.01)	0.046	−0.20 (0.04)	−0.17 (0.01)	0.423
Niacin, mg	16.16 (0.10)	13.2 (0.10)	15.40 (0.59)	13.12 (0.10)	<0.001	−3.55 (0.49)	−2.94 (0.10)	0.241
Vitamin B6, mg	2.16 (0.01)	1.71 (0.01)	1.84 (0.06)	1.71 (0.01)	0.016	−0.58 (0.06)	−0.44 (0.01)	0.026
Folic acid, ug	326.17 (1.58)	249.18 (1.70)	241.35 (8.98)	249.45 (1.73)	0.392	−110.01 (9.68)	−75.85 (1.96)	<0.001
Vitamin E, mg	15.7 (0.09)	13.2 (0.09)	14.08 (0.51)	13.17 (0.09)	0.077	−3.09 (0.50)	−2.48 (0.10)	0.278
Vitamin C, mg	151.47 (1.58)	113.10 (1.55)	96.33 (7.56)	113.68 (2.58)	0.044	−62.64 (10.67)	−37.53 (1.87)	0.015
Protein, g	73.49 (0.39)	62.26 (0.39)	71.32 (2.63)	61.94 (0.40)	<0.001	−12.50 (2.10)	−11.19 (1.87)	0.531
Carbohydrate, g	260.46 (1.31)	228.21 (1.37)	238.23 (7.13)	227.87 (1.40)	0.175	−33.16 (7.90)	−32.22 (1.45)	0.906
Lipid, g	50.3 (0.35)	45.58 (0.36)	53.26 (2.83)	45.31(0.36)	0.008	−4.90 (1.91)	−4.72 (0.36)	0.930
Cholesterol, mg	317.55 (3.84)	336.11 (2.57)	367.52 (23.68)	335.03 (3.88)	0.128	20.57 (19.78)	18.49 (3.96)	0.923
Total calories	1818.37 (6.77)	1766.56 (6.32)	1948.73 (32.52)	1760.27 (6.37)	<0.001	−71.49 (9.08)	−51.13 (1.56)	0.018

* Compared between groups with and without a new airflow limitation. All values are expressed as the mean (standard error) uptake per day. RE, retinol equivalents.

## Data Availability

Individual participant data collected during the study, after de-identification, and study protocols and statistical analysis code are available beginning 3 months and ending 1 years following article publication to researchers who provide a methodological sound proposal, with approval by an independent review committee (“learned intermediatry”) identified for purpose. Data is available for analysis to achieve aims in the approved proposal. Proposals should be directed to seiwon@amc.seoul.kr; to gain access, data requestors will need to sign a data access or material transfer agreement approved by Asan Medical Center.

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
