# Peer review of "The Impact of Changes in the Intake of Fiber and Antioxidants on the Development of Chronic Obstructive Pulmonary Disease"

_nutrients, 2021, doi:10.3390/nu13020580_

Round 1
Reviewer 1 Report
In this study by Young Ju Jung et al, entitled “The Impact of Changes in the Intake of Fiber and Antioxidants 2 on the Development of Airflow Limitation” the authors investigated the role of diet in COPD progression and development, showed that a decreased intake of dietary fiber, vitamin C, and folic acid is associated with a newly developed airflow limitation.
The manuscript resulted well written and clear, also the selection of the population and the study design were clearly reported.
Some point can be improved:
- Specify in the title that with Airflow Limitation you referred to COPD
- The aim of the study and the principal results obtained need to be added in the aim of the study.
- The table 1 is not clear. Please report all p-value or report only significant p value.
- Better explain the significance of WBC in lung disease that in the last year seemed to be relevant for the development of lung diseases. (Bergantini L, d'Alessandro M, Vietri L, Rana GD, Cameli P, Acerra S, Sestini P, Bargagli E. Utility of serological biomarker' panels for diagnostic accuracy of interstitial lung diseases. Immunol Res. 2020 Dec;68(6):414-421. doi: 10.1007/s12026-020-09158-0. Epub 2020 Oct 22. PMID: 33089426; PMCID: PMC7674352.)
Author Response
Review1
In this study by Young Ju Jung et al, entitled “The Impact of Changes in the Intake of Fiber and Antioxidants 2 on the Development of Airflow Limitation” the authors investigated the role of diet in COPD progression and development, showed that a decreased intake of dietary fiber, vitamin C, and folic acid is associated with a newly developed airflow limitation.
The manuscript resulted well written and clear, also the selection of the population and the study design were clearly reported.
Some point can be improved:
Specify in the title that with Airflow Limitation you referred to COPD
→ We deeply appreciate your comments. The title of this manuscript was changed as; The Impact of Changes in the Intake of Fiber and Antioxidants on the Development of Chronic Obstructive Pulmonary Disease
The aim of the study and the principal results obtained need to be added to the aim of the study.
→ We agreed with your comment, so the below sentences were added to lines 54-55 on page 3 of the introduction section.
We hypothesized that the dietary intake includes dynamic changes that would have an impact on the airflow limitation, which is the principal feature of COPD, and each nutrient might be associated with it
Table 1 is not clear. Please report all p-value or report only significant p-value.
→ According to the review’s comment, the p-value not filled in table 1 was added. Please check table 1.
Better explain the significance of WBC in lung disease that in the last year seemed to be relevant for the development of lung diseases. (Bergantini L, d'Alessandro M, Vietri L, Rana GD, Cameli P, Acerra S, Sestini P, Bargagli E. Utility of serological biomarker' panels for diagnostic accuracy of interstitial lung diseases. Immunol Res. 2020 Dec;68(6):414-421. DOI: 10.1007/s12026-020-09158-0. Epub 2020 Oct 22. PMID: 33089426; PMCID: PMC7674352.)
→ According to the review’s comment, the below sentence and references were added to lines 218 on page 10 of the discussion section.
In our present analyses, WBC counts, an inflammatory marker, were associated with reduced lung function [32, 33] and were affected by the intake of fiber in some of the subjects.
Reference>
- Gan W.Q.; Man S.F.; Senthilselvan A.; Sin D.D. Association between chronic obstructive pulmonary disease and systemic inflammation: a systematic review and a meta-analysis. Thorax. 2004, 59, 574-580
- Yang H.F.; Kao T.W.; Wang C.C.; Peng T.C.; Chang Y.W.; Chen W.L. Serum white blood cell count and pulmonary function test are negatively associated. Acta Clin Belg. 2015, 70, 419-424.
Reviewer 2 Report
A cohort of 1439 healthy participants were followed for 5 years with changes in lung function, biochemistry and dietary intake assessed.
Of these 3.3% (n=48) developed a small degree of airflow limitation. Statistical analysis showed that these changes were associated with a reduced dietary intake of fibre and Vitamin C.
The study is well presented and fully supports the conclusions provided. The strength of the study is the size of the sample population.
A greater difference may have become apparent if an older population were monitored. Overall a worthy and interesting study.
Author Response
Review2
A cohort of 1439 healthy participants were followed for 5 years with changes in lung function, biochemistry and dietary intake assessed.
Of these 3.3% (n=48) developed a small degree of airflow limitation. Statistical analysis showed that these changes were associated with a reduced dietary intake of fibre and Vitamin C.
The study is well presented and fully supports the conclusions provided. The strength of the study is the size of the sample population.
A greater difference may have become apparent if an older population were monitored. Overall a worthy and interesting study.
→ We deeply appreciate your comments. Maintaining the intake of fiber and vitamins in older people is also becoming an important issue in ageing societies. A decreased intake of fiber and antioxidants might be particularly important for the older population who are susceptible to lung decline. A future interventional study with detailed monitoring of lifestyles may provide clearer findings.
